# Selenate Prevents Adipogenesis through Induction of Selenoprotein S and Attenuation of Endoplasmic Reticulum Stress

**DOI:** 10.3390/molecules23112882

**Published:** 2018-11-05

**Authors:** Choon Young Kim, Kee-Hong Kim

**Affiliations:** 1Department of Food and Nutrition, Yeungnam University, 280 Daehak-ro, Gyeongsan, Gyeongbuk 38541, Korea; 2Department of Food Science, Purdue University, West Lafayette, IN 47897, USA; keehong@purdue.edu

**Keywords:** selenium, selenate, selenoprotein S, ER stress, adipogenesis, 3T3-L1 preadipocytes

## Abstract

The conversion of preadipocytes to adipocytes (adipogenesis) is a potential target to treat or prevent obesity. Selenate, an inorganic form of selenium, elicits diverse health benefits, mainly through its incorporation into selenoproteins. The individual roles of selenium and certain selenoproteins have been reported. However, the effects of selenate treatment on selenoproteins in adipocytes are unclear. In this study, the effects of selenate pretreatment on selenoprotein and endoplasmic reticulum (ER) stress during adipogenesis were examined in vitro. The selenate pretreatment dose-dependently suppressed the adipogenesis of 3T3-L1 preadipocytes. The selenate pretreatment at 50 μM for 24 h almost completely suppressed adipogenesis without cytotoxic effects. The expression of the adipogenic genes *peroxisome proliferator-activated receptor gamma, CCAAT-enhancer binding protein alpha*, and *leptin* was suppressed by selenate. This pretreatment also upregulated selenoprotein S (SEPS1), an ER resident selenoprotein that reduces ER stress, and prevented dexamethasone-induced SEPS1 degradation during the early stage of adipogenesis. The selenate-inhibited adipogenesis was associated with an attenuation of ER stress. The expression of the ER stress marker genes was upregulated during the early stage of differentiation, whereas the selenate pretreatment suppressed the mRNA expression of the XBP1 and C/EBP homologous protein. The collective data suggest a preventive role of selenate and SEPS1 in adipogenesis, and support a novel dietary approach to prevent obesity.

## 1. Introduction

Adipocytes are potent endocrine cells that synthesize and secrete a large amount of protein and lipid mediators [1]. Adipogenesis, which is adipocyte hyperplasia, is associated with an increased adipose tissue mass in obesity. During this process, the cells undergo dramatic biochemical and morphological changes, and are overwhelmed by newly produced proteins. The endoplasmic reticulum (ER) is the intracellular organelle for the synthesis, folding, and modification of secretory and cell-surface proteins, as well as lipid biosynthesis. ER also maintains its homeostasis by increasing the chaperone-mediated protein folding capacity, inhibiting general protein synthesis, and promoting misfolded protein degradation through ER-associated protein degradation (ERAD).

ER stress, defined as the accumulation of misfolded proteins within the ER, initiates the unfolded protein response [2]. ER stress is sensed by three ER-resident transmembrane proteins, inositol-requiring protein 1 (IRE1), PKR (double-stranded-RNA-dependent protein kinase)-like ER kinase (PERK), and activating transcription factor 6 (ATF6). Glucose-regulated proteins (GRP or BiP)/chaperones are induced in response to ER stress [3]. While GRP, the most abundant chaperone in the ER lumen, binds to IRE1, PERK, and ATF6 in non-stressed cells, the GRP chaperone is dissociated from three ER sensor proteins upon the activation of unfolded protein response (UPR), and binds to the unfolded protein that is to be removed. Free IRE1 catalyzes mRNA of the X-box-binding protein (XBP1), producing s-XBP1, which is translated from the spliced XBP1 mRNA and, activates the transcription of ER-stress-related genes like GRP78. Free PERK is homodimerized and autophosphorylated to phosphorylated eukaryotic initiation factor 2 alpha (eIF2α), which subsequently inhibits the assembly of the 80S ribosome. This results in a general inhibition of the protein translation and expression of the pro-apoptotic transcription factor C/EBP homologous protein (CHOP). ATF6 moves to the Golgi complex and undergoes proteolytic cleavage. The cleaved ATF6 is translocated from the cytoplasm to the nucleus, resulting in the up-regulation of the transcription of genes, such as CHOP, GRP78, and XBP1.

ER stress is positively correlated with body mass index and obesity, as cellular events occurring during adipogenesis confer ER stress in the cells [4,5]. Increased ER stress is found in diet-induced obesity and genetically obese models. Furthermore, a high level of ER stress has been associated with insulin resistance in obese animals [4]. GRP78 heterozygosity attenuates diet-induced obesity and insulin resistance [1]. Among the ER stress sensors, the IRE1-XBP1 pathway is an essential branch for adipogenesis. The early adipogenic transcription factor C/EBPβ activates C/EBPα through the IRE1α-XBP1 pathway [6]. XBP1 is also critical for lipogenesis in the liver [7]. Interestingly, ER stress is decreased in the liver and adipose tissues of obese subjects after weight loss [8]. A reduction in ER stress by the chemical chaperones, 4-phenyl butyric acid and tauroursodeoxycholic acid, effectively inhibits body weight gain and adipocyte differentiation [9,10].

Selenium is an essential micronutrient with potent anti-oxidative and anti-inflammatory properties, as well as estrogenicity [11,12]. It has been extensively studied as a potential agent to prevent various types of cancer and cardiovascular diseases [11,13]. A U-shaped non-linear dose–response relationship between selenium supplementation and type 2 diabetes has been reported [14], and the role of selenium in the development of obesity was recently reported. In vivo animals that were fed a selenium supplementation exhibited less body weight gain [15]. Selenate, an inorganic form of selenium, inhibits high-fat diet-induced obesity in C57BL/6J mice [16] and adipogenesis through the induction of transforming growth factor-beta (TGF- β) signaling [17].

The variety of reported beneficial health effects of selenium occur mainly through its incorporation into selenoproteins [18]. SEPS1 (also termed selenoprotein S, SelS, Tanis, or VCP-interaction membrane protein) is a selenoprotein localized in the plasma membrane [19] and ER membrane [20]. Intracellular SEPS1 controls ER stress. SEPS1 is a component of the retrotranslocation/ERAD machinery, which controls ER homeostasis [21,22]. Cytotoxicity and apoptosis caused by the ER stress inducer are prevented by SEPS1 in macrophages [23]. The SEPS1 expression is induced by glucose deprivation and ER stress [24,25]. Recently, the role of SEPS1 in adipogenesis has been reported. During the early stage of differentiation, SEPS1 degradation is necessary for 3T3-L1 adipogenesis [26,27].

Even though the individual roles of selenate and SEPS1 in adipogenesis are known, the effect of selenate pretreatment on SEPS1 during adipogenesis is unclear. The objective of this study was to determine the role of selenate pretreatment in the SEPS1 expression, adipogenesis, and ER stress of 3T3-L1 preadipocytes during adipogenesis.

## 2. Results

### 2.1. Selenate Pretreatment Suppresses Adipogenesis in 3T3-L1 Preadipocytes

To examine the effect of selenate pretreatment on adipogenesis, 50 µM of selenate was added for various incubation times, prior to the initiation of differentiation (Day 0). Oil Red O staining was performed on Day 6 (Figure 1A). The selenate lowered the adipogenesis of 3T3-L1 adipocytes in a time-dependent manner. Pretreatment for 24 h was required in order to observe almost complete inhibition (Figure 1B). Therefore, selenate pretreatment for 24 h was used in the following experiments.

To determine whether the lower level of lipid accumulation observed in 3T3-L1 cells pretreated with selenate reflected toxicity, the viability and cytotoxicity of selenate-treated 3T3-L1 preadipocytes were measured using the 3-(4,5-Dimethylthiazol-2-yl)-2,5-diphenyltetrazolium bromide (MTT) and lactate dehydrogenase (LDH) activity assays, respectively. Treatment with 0, 10, 25, 50, and 100 µM selenate for 24 h had no significant effect on the viability or cytotoxicity of preadipocytes (data not shown.). Thus, the anti-adipogenic function of selenate was independent of selenate-induced cytotoxicity. Other cellular events specifically induced by selenate might be involved in its anti-adipogenic effect.

### 2.2. Selenate Pretreatment Inhibits PPARγ and C/EBPα Gene Expression in the Early Phase of Adipogenesis

As the expression of the early adipogenic transcriptional factors is critical for the progression of adipogenesis, we determined the mRNA expression of early adipogenic transcriptional factor genes during the early stage of adipogenesis in selenate-pretreated adipocytes using quantitative RT-PCR. On Day 2, a 24 h pretreatment with 50 µM selenate did not affect the *C/EBPβ* gene expression (Figure 2A), but it significantly inhibited the mRNA expression of the adipogenic transcription factors, *PPARγ* and *C/EBPα* (Figure 2B,C). Another adipocyte marker gene, *leptin*, was also significantly reduced by the selenate pretreatment at Day 2 (Figure 2D). It is notable that only 24 h of selenate pretreatment prior to the initiation of differentiation suppressed the expressions of *PPARγ* and *C/EBPα*.

As the activation of the PPARγ is required for adipogenesis, and selenate pretreatment suppresses the *PPARγ* gene expression, we tested the effect of the PPARγ ligand rosiglitazone on the reversal of the preventive action of selenate [28]. One-day post-confluent 3T3-L1 preadipocytes were treated with selenate (0, 25, and 50 µM) for 24 h, prior to the adipogenesis of the cells in the presence and absence of 1 µM rosiglitazone for two days. On Day 6, the intracellular lipid accumulation of these cells was assessed. The selenate pretreatment dose-dependently suppressed the adipocyte differentiation (Figure 3A). The quantification of Oil Red O staining revealed a decrease in the adipogenesis by 30% using 25 µM selenite, and a near-total inhibition of adipogenesis with 50 µM selenite (Figure 3B). The administration of rosiglitazone reversed the inhibition of adipogenesis in the selenate-treated cells (Figure 3).

### 2.3. Regulation of SEPS1 Protein by Selenium and Dexamethasone Treatment

As only a 24 h selenate pretreatment prior to the initiation of adipogenesis was able to work for the six days of differentiation, it was reasonable to expect that there may be intermediates. The selenate treatment may induce selenoproteins in 3T3-L1 preadipocytes. As SEPS1 protein is involved in adipogenesis, the gene and protein levels of SEPS1 were determined. During the early stage of differentiation, the *SEPS1* mRNA level was elevated on Day 2, but the SEPS1 protein level was oppositely decreased (Figure 4). In contrast, decreased SEPS1 protein levels during the early stage of adipogenesis were prevented by selenate pretreatment, but the mRNA expression of *SEPS1* was not changed by the selenate addition (Figure 4).

To study what affects the SEPS1 protein expression during adipogenesis, the effects of selenate and each component of the adipogenic cocktail on the SEPS1 expression were determined. Among the adipogenic hormones, dexamethasone (DEX) was able to decrease the levels of the SEPS1 protein, but the selenate pretreatment blocked the DEX-induced SEPS1 protein degradation (Figure 5A). On Day 0 and 1, the cells treated with selenate for 24 h prior to adipogenesis initiation showed elevated levels of SEPS1 protein, compared to the control (Figure 5B). Moreover, the SEPS1 level was dramatically reduced at Day 1, but this reduction was reversed by treatment with the proteasome inhibitor MG132, suggesting an association of degradation of the SEPS1 protein with proteasomal activity (Figure 5B). These results indicated that the SEPS1 degradation during the adipogenesis is prevented by selenate pretreatment.

### 2.4. Selenate Pretreatment Inhibits ER Stress Marker Gene Expression

ER stress is one of the earliest events activated during adipogenesis. As selenium and SEPS1 have previously been reported to mediate ER stress, we examined the effect of selenate on ER stress and SEPS1 expression in the early phase of adipogenesis. The mRNA expression of the ER stress marker genes was measured using quantitative RT-PCR. On Day 2, the expressions of ER stress marker genes, *XBP1*, (*GRP78*, and *CHOP*, were significantly suppressed by selenate pretreatment (Figure 6A–C). Consistent with this result, the phosphorylation of eukaryotic initiation factor 2 alpha (eIF2α) and protein level of GRP78 were inhibited by pretreatment with 50 µM selenate on Day 1 and 2 (Figure 6D).

To confirm the inhibitory role of selenate in ER stress, the chemical ER-inducer tunicamycin was added to 3T3-L1 preadipocyte culture in the absence and presence of selenate for 6 h. Gene expression analysis revealed that tunicamycin treatment significantly increased mRNA expression of *XBP1*, *GRP78*, *CHOP*, and *XBP1*-downstream target genes, *ER degradation enhancing alpha-mannosidase-like protein* (*EDEM*) and *protein-disulfide isomerase-associated 3* (*Pdia3*) (Figure 7A–E). Tunicamycin-induced ER stress marker gene expression was suppressed by selenate treatment in a dose-dependent manner (Figure 7). Therefore, selenate has the ability to reduce tunicamycin-induced ER stress in 3T3-L1 preadipocytes. These findings supported the idea that selenate pretreatment attenuates ER stress in 3T3-L1 preadipocytes, which could occur through the up-regulation of SEPS1 protein.

## 3. Discussion

Obesity is a major threat to public health. Controlling obesity by dietary components, such as micronutrients, has been exclusively studied [29]. 3T3-L1 preadipocytes are well-established in vitro models used to study the regulation of obesity. Transcription factors, including C/EBPβ, PPARγ, and C/EBPα, and ER stress are important inducers of adipogenesis program [30,31]. Selenium has a variety of health beneficial actions. Selenium supplementation inhibits high fat diet-induced obesity [16] and adipogenesis by activating of TGF-1β signaling [17]. However, there is scant evidence of the role of selenoproteins induced by selenium in adipogenesis.Herein, we investigated the role of selenate and SEPS1 in adipogenesis, and sought to clarify the underlying mechanisms during the differentiation of 3T3-L1 preadipocytes.

Selenate pretreatment prior to the initiation of adipogenesis suppressed 3T3-L1 differentiation. Pretreatment with 50 µM selenate for at least 24 h was required in order to induce the maximum inhibition of lipid accumulation (Figure 1 and Figure 3). The results of the cell viability and cytotoxicity assays supported the safety of using selenate at concentrations up to 100 µM (data not shown). Therefore, the inhibition of lipid accumulation induced by selenate does not occur, because of selenium-related toxicity, but rather through selenate-dependent unknown mechanisms. To elucidate the molecular pathway underlying the action of selenate, we examined possible cellular signaling pathways that could be critical for adipogenesis, and estimated the expression of important early adipogenic transcription factors. Quantitative RT-PCR revealed that the levels of *PPARγ* and *C/EBPα* mRNA were significantly suppressed by the selenate pretreatment on Day 2, while their direct upstream *C/EBPβ* gene expression was not influenced (Figure 2). This result implicated that selenate might modulate the expression of C/EBPβ and its downstream genes during the early stage of adipogenesis. Supporting to this motion, the IRE-XBP1 pathway of ER sensing has been reported to be essential for adipogenesis, because the IRE1α-XBP1 pathway is regulated by C/EBPβ, resulting in the activation of C/EBPα [6]. Moreover, we previously reported the action of selenate during the course of adipogenesis [17]. Our previous study showed that a 6-days of selenate treatment during the course of adipogenesis resulted in inhibition of adipogenesis through the induction of TGF-1β signaling [17].

Our current study showed that selenate pretreatment only for 24 h prior to the adipogenesis was able to inhibit adipogenesis, indicating that there might be an intermediate in the regulation of adipogenesis. As selenium functions through incorporation into selenoproteins [18], we wanted to identify the selenoprotein that is critical for the action of selenate on adipogenesis. SEPS1 is known to modulate ER stress and adipogenesis [24,26,27], we investigated how SEPS1 is regulated by selenate pretreatment in adipogenesis. The SEPS1 protein level was dramatically decreased by dexamethasone (DEX) during adipogenesis. However, the DEX-induced SEPS1 degradation was prevented by the selenate pretreatment during the early stage of differentiation (Figure 4 and Figure 5). Interestingly, selenate did not affect the *SEPS1* mRNA expression, providing evidence that selenate influences the SEPS1 level following translation. We previously reported that DEX is able to decrease the levels of SEPS1 protein through proteasome degradation, and that the degradation of SEPS1 is required for adipogenesis [27]. Thus, the sustained SEPS1 protein expression due to selenate pretreatment prevents lipid accumulation. In addition to SEPS1, other selenoproteins are also reported to modulate. The selenoprotein P knockdown inhibits adipogenesis through the induction of the inflammatory response [32]. Recently, an inhibitory effect of selenoprotein K, together with SEPS1 on adipogenesis, has been reported to be involved in adipogenesis [26].

Given a critical role of ER stress in adipogenesis a protective role of SEPS1 in ER stress-induced cellular signaling pathway [23], we investigated the effect of selenate pretreatment on ER stress during the early stage of differentiation. On Day 2, the expressions of ER stress markers were significantly increased, while selenate pretreatment reduced this up-regulation (Figure 6). Selenate also effectively suppressed tunicamycin-induced ER stress (Figure 7). As the upregulated levels of GRP78, phosho-eIF2α, and XBP1 are essential for adipogenesis [9], the suppression of these ER stress markers by selenate pretreatment may contribute to the suppression of adipogenesis. Therefore, selenate pretreatment may synergistically prevent adipogenesis through the inhibition of SEPS1 protein degradation by DEX, and the suppression of expression of adipogenic transcription factors and ER stress markers during the early stage of adipogenesis.

Overall, our results show preventive potential of selenate in adipogenesis by attenuating ER stress inducing SEPS1 during the early stage of differentiation. Further research is necessary to demonstrate the physiological relevance of these findings in vivo. If validated in vivo, the present findings could provide valuable data for a novel dietary approach to prevent obesity. Additionally, our findings provide evidence-based rational for food industry to develop or screen healthier foods (or food ingredients) enriched with specific forms of selenium (i.e., selenate).

## 4. Materials and Methods

### 4.1. Materials and Reagents

The DEX, IBMX, insulin, tunicamycin, MG132, and Oil Red O powder were purchased from Sigma-Aldrich (St. Louis, MO, USA). The fetal calf serum (FCS) and fetal bovine serum (FBS) were purchased from the PAA Cell Culture Company (Worcester, MA, USA). The Dulbecco’s Modified Eagle’s medium (DMEM) and 0.25% trypsin-EDTA were obtained from Thermo Fisher Scientific (Waltham, MA, USA). The rosiglitazone was purchased from Calbiochem (San Diego, CA, USA). The selenate, dimethyl sulfoxide, and 3-(4,5-dimethyl-thiazol-yl-2)-2,5-diphenyltetrazolium bromide were purchased from Alfa Aesar (Ward Hill, MA, USA).

### 4.2. Cell Culture and Differentiations of 3T3-L1 Preadipocytes

The 3T3-L1 preadipocyte cell line was obtained from the American Type Culture Collection (Manassa, VA, USA). The preadipocytes were maintained in DMEM containing 10% (*v*/*v*) FCS. Differentiation was initiated in 3T3-L1 preadipocytes two days after achieving confluent growth (designated as Day 0) using DMEM supplemented with 10% FBS and an adipogenic cocktail including 167 nM insulin, 0.5 mM IBMX, and 5 μM DEX. The cocktail was supplied for two days. The cells were then cultured in 10% FBS-DMEM containing insulin for another two days, followed by an additional two days of culture with 10% FBS-DMEM. At that time, >90% of the cells had differentiated into mature adipocytes. All of the media additionally contained 100 U/mL penicillin, 100 μg/mL streptomycin, and sodium pyruvate. The cells were maintained at 37 °C in a humidified 5% CO_2_ atmosphere. After Oil Red O staining was performed on Day 6, the stained cells were visualized by light microscopy and were photographed. The lipid accumulation was determined by measuring the optimal density of the stained lipids, 490 nm after extracting the dye from cells using isopropanol.

### 4.3. Isolation of Total RNA and Quantitative RT-PCR

The total RNA in the 3T3-L1 preadipocytes was extracted using Trizol^®^ Reagent (Invitrogen, Carlsbad, CA, USA), according to the manufacturer’s instruction. For the cDNA synthesis, 1 μg of isolated RNA was subjected to a reverse transcription reaction using the SuperScriptII kit (Invitrogen, Carlsbad, CA, USA). The generated cDNA was subjected to quantitative RT-PCR using a thermocyclers purchased from Applied Biosystems (Carlsbad, CA, USA). The sequences of the primers corresponding to the murine adipogenic genes for quantitative RT-PCR were as follows: *CCAAT/enhancer-binding protein beta (C/EBPβ)* (forward, 5′-AGC GGC TGC AGA AGA AGG T-3′; reverse, 5′-GGC AGC TGC TTG AAC AAG TTC-3′); *peroxisome proliferator-activated receptor gamma (PPARγ)* (forward, 5′-CCC AAT GGT TGC TGA TTA CAA AT-3′; reverse, 5′-CTA CTT TGA TCG CAC TTT GGT ATT CT-3′); *C/EBPα* (forward, 5′-GGT TTA GGG ATG TTT GGG TTT TT-3′; reverse, 5′-AAG CCC ACT TCA TTT CAT TGG T-3′); *leptin* (forward, 5′-CAC ACA CGC AGT CGG TAT CC-3′; reverse, 5′- AGC CCA GGA ATG AAG TCC AA-3′); *SEPS1* (forward, 5′-GCT GGA CCA AGC CGA GAC T-3′; reverse, 5′-CTA GAA GCA TTT GCG GTG GAC GAT GGA GGG-3′); *XBP-1* (forward, 5′-GGA TTT GGA AGA AGA GAA CCA CAA-3′; reverse, 5′-CCG TGA GTT TTC TCC CGT AAA A-3′); *glucose-regulated protein78 (GRP78)* (forward, 5′-CGG ACG CAC TTG GAA TGA C-3′; reverse, 5′-AAC CAC CTT GAA TGG CAA GAA-3′); *C/EBP homologous protein (CHOP)* (forward, 5′-GTC CTG TCC TCA GAT GAA ATT GG-3′; reverse, 5′-AAG GTT TTT GAT TCT TCC TCT TCG T-3′); and β*–actin* (forward, 5′-TGA CGG GGT CAC CCA CAC TGT GCC CAT CTA-3′; reverse, 5′-CTA GAA GCA TTT GCG GTG GAC GAT GGA GGG-3′).

### 4.4. Immunoblot Analysis

Differentiation was initiated in the 3T3-L1 cells that had achieved confluent growth for two days using a standard differentiation cocktail in the absence or presence of selenate (0 and 50 µM). The cells were harvested by scrapping at Day 0, 1, and 2, and were collected by centrifugation. The cells were lysed in a buffer containing Tris-Cl (100 mM, pH 7.5–8.0), NaCl (100 mM), 0.5% TritonX-100, protease inhibitor cocktail, sodium orthovanadate (1 mM), and sodium fluoride (10 mM). The protein concentration was estimated by the Bradford method (Bio-Rad Laboratories, Hercules, CA, USA). The proteins were separated by 10% SDS-PAGE and were transferred to a nitrocellulose membrane. Immunoblot was performed by anti-VIMP (anti-SEPS1; Sigma-Aldrich, St. Louis, MO, USA), anti-BiP (anti-GRP78; GenScript, Piscataway, NJ, USA), p-eIF2α (Cell Signaling Technology, Danvers, MA, USA), and anti-β-actin (Santa Cruz Biotechnology, Santa Cruz, CA, USA) antibodies at 4 °C overnight. The binding of each primary antibody was visualized with horseradish peroxidase-conjugated secondary antibodies (Santa Cruz Biotechnology, Dallas, TX, USA), using an enhanced chemiluminescence kit (Pierce, Rockford, IL, USA).

### 4.5. Statistical Analysis

The data are shown as mean ± standard error of mean (SEM). The statistical analysis was performed using SAS9.2 software (SAS Institute, Inc., Cary, NC, USA). One-way ANOVA were used to determine significance of treatment effect and interactions. Significant differences between the group means were assessed using Dunnett’s multiple comparison, and significance was indicated at *p* < 0.05.

## Figures and Tables

**Figure 1 molecules-23-02882-f001:**
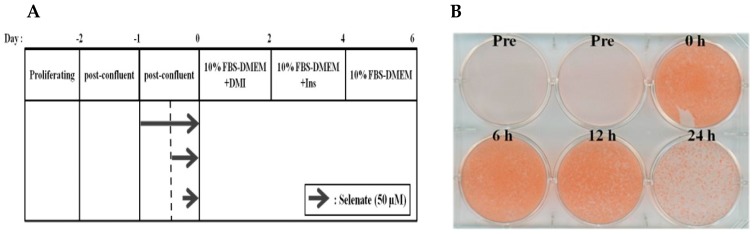
Selenate pretreatment inhibits adipogenesis of 3T3-L1 cells. (**A**) One-day post-confluent 3T3-L1 preadipocytes (Pre) were pretreated 50 µM selenate for 0, 6, 12, and 24 h, and were differentiated until Day 6 without selenate treatment. (**B**) Oil Red O staining of cells was done to determine the accumulated lipid droplets.

**Figure 2 molecules-23-02882-f002:**
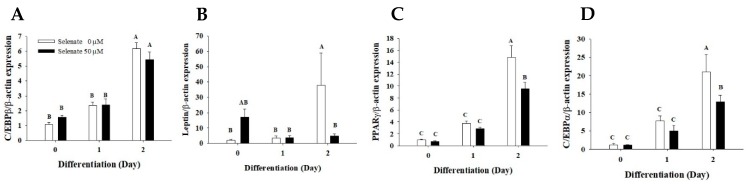
Selenate pretreatment suppresses the *PPARγ* and *C/EBPα* gene expression during the early stage of differentiation. After 24 h pretreatment of one day post-confluent 3T3-L1 preadipocytes with 50 µM selenate, the cells were differentiated. RNAs were collected at Day 0, 1, and 2 during adipogenesis. Quantitative RT-PCR was performed to estimate the mRNA expression of the adipogenic transcription factors, (**A**) *C/EBPβ*, (**B**) *PPARγ*, and (**C**) *C/EBPα*, and the adipogenic marker gene (**D**) *leptin*. The data represents mean ± standard error of mean (SEM). The experiment was conducted three times (*n* = 12). Different characters indicate a significant difference at *p* < 0.05.

**Figure 3 molecules-23-02882-f003:**
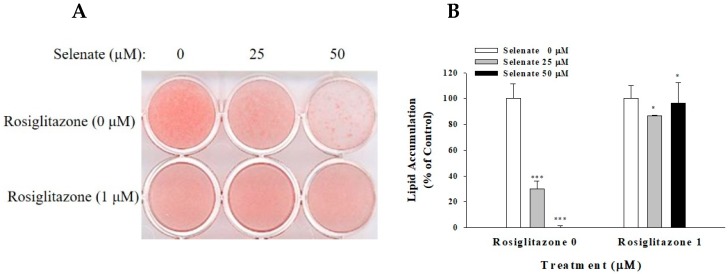
Rosiglitazone abolishes selenate-inhibited adipogenesis. After a 24 h pretreatment of one-day post-confluent 3T3-L1 preadipocytes with 50 µM of selenate, the cell differentiation was initiated by the addition of an adipogenic cocktail with 0 or 1 µM rosiglitazone during Day 0–2, in the absence of selenate. On Day 2, the cells were transferred to Dulbecco’s modified Eagle’s medium (DMEM) containing 10% fetal bovine serum (FBS-DMEM) and insulin, and to 10% FBS-DMEM on Day 4–6. At Day 6, Oil Red O staining was performed (**A**). The quantitative analysis of staining is shown in (**B**). The data represent mean ± SEM. The experiment was conducted three times (*n* = 3–9). * *p* < 0.05; *** *p* < 0.001.

**Figure 4 molecules-23-02882-f004:**
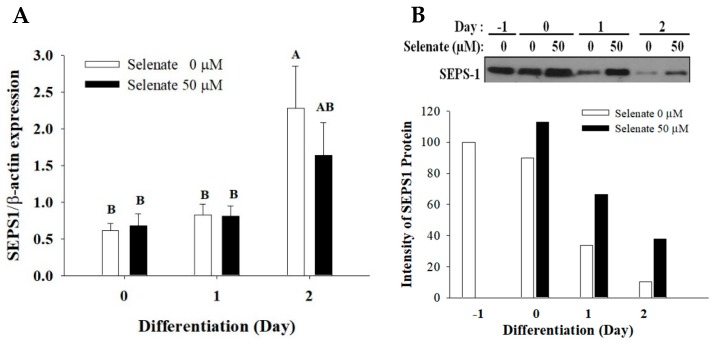
Selenate pretreatment induces a selenoprotein S (SEPS1) protein expression, but not a gene expression. After a 24 h pretreatment of one-day post-confluent 3T3-L1 preadipocytes with 50 µM selenate, the cells were differentiated during Day 0–2 in the absence of selenate. During the early stage of differentiation, the SEPS1 gene (**A**) and protein (**B**) levels were determined by quantitative RT-PCR and immunoblot analysis, respectively. The data represent mean ± SEM. The experiment was conducted three times (*n* = 3 or 9). Different characters indicate a significant difference at *p* < 0.05.

**Figure 5 molecules-23-02882-f005:**
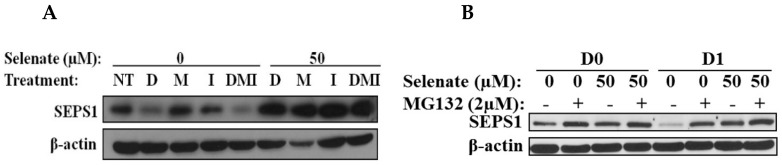
Regulation of SEPS1 by selenate and adipogenic cocktail treatments. (**A**) After 24 h pretreatment of one-day post-confluent 3T3-L1 preadipocytes with 50 µM selenate, the cells were differentiated in individual components of the adipogenic cocktail, dexamethasone (D), isobutylmethylxanthine (M), and insulin (I), and DMI. Protein samples were collected on Day 2 (D2) and were subjected to immunoblot analysis. (**B**) The samples shown in D0 are one-day post-confluent 3T3-L1 preadipocytes added in the presence or absence of selenate and MG132 for 24 h. The samples shown in D1 include 3T3-L1 preadipocytes treated with DMI with or without MG132 for 24 h, after selenate pretreatment for 24 h. The protein samples at D0 and D1 were subjected to immunoblot assay.

**Figure 6 molecules-23-02882-f006:**
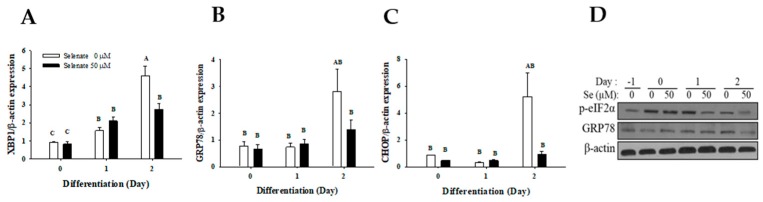
Selenate pretreatment inhibits endoplasmic reticulum (ER) stress marker gene expression during the early stage of adipogenesis. After 24 h pretreatment with 50 µM selenate (Se), RNA and protein samples were collected at Day 0, 1, and 2 of differentiation. Quantitative RT-PCR was performed to measure mRNA expression of (**A**) *XBP1*, (**B**) *GRP78*, and (**C**) *CHOP*. (**D**) Protein levels of p-eIF2α and GRP78 were assessed by immunoblot analysis. Data represents mean ± SEM (*n* = 12). β-actin was an internal standard. Data represents mean ± SEM (*n* = 12). Different characters indicate significant difference at *p* < 0.05.

**Figure 7 molecules-23-02882-f007:**
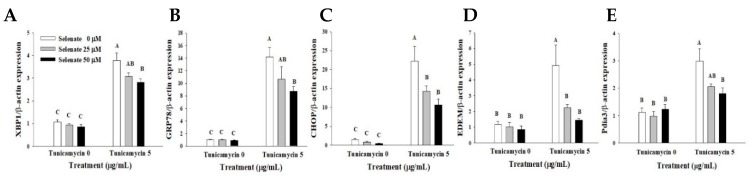
Selenate treatment suppresses tunicamycin-induced endoplasmic reticulum (ER) stress. After 24 h of pretreatment with 25 and 50 µM selenate in one-day post-confluent 3T3-L1 preadipocytes, the cells initiated differentiation during Day 0–2 in the absence of selenate. On Day 2, the levels of ER-stress related genes *XBP-1* (**A**), *GRP78* (**B**), *CHOP* (**C**), *EDEM* (**D**), and *Pdia3* (**E**) had their levels determined using quantitative RT-PCR. Different characters indicate a significant difference at *p* < 0.05.

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
