# Peer review of "Selenate Prevents Adipogenesis through Induction of Selenoprotein S and Attenuation of Endoplasmic Reticulum Stress"

_molecules, 2018, doi:10.3390/molecules23112882_

Round 1
Reviewer 1 Report
The authors have identified a novel role for selenate on adipogenic differentiation through SEPS/ER stress/PPARgamma axis. Activation of PPARgamma by rosiglitazone bypasses this axis and leads into adipogenesis even in the presence of selenite suggesting that PPARgamma activation exhibits powerful role against this action of selenite in preadipocytes. The studies are well designed and the results are supporting the author's conclusions. Mechanisms are well explained and the authors have taken tremendous efforts to fully understand the role of selenite on adipogenesis that could impact the obesity and associated complications. However, since obesity is associated with abnormalities in lipid metabolism, it would be additionally beneficial if the authors could establish if the selenite can also act on mature adipocytes to modulate the metabolic functions of mature adipocytes: glucose uptake and lipolysis. As ER stress can downregulate the action of insulin on adipocyte functions, it would be beneficial to understand if selenite can also impact ER stress in mature adipocytes in influencing one or more of its functions. Other than this one experiment, the study is complete in all aspects and the results will create a huge impact to the area of adipogenesis and obesity.
Author Response
The reviewer pointed out very important research area on the function of selenate in adipocytes that we have not studied. Even though current study only elucidated the effect of selenate on selenoprotin S expression and adipogenesis in preadipocytes, the regulation of metabolic function by selenate in mature adipocyte will be investigated in the next study. We appreciate your valuable suggestion.
Reviewer 2 Report
The manuscript, molecules-382094,describes that selenate prevents adipogenesis through induction of selenoprotein S and attenuation of ER stress. It is interesting that inorganic compound present prevents adipogenesis with reasonable experiments for support of the author’s discussion. In this time, this manuscript is suitable to publish Molecule after the minor revision.
The reviewer consider that it is not enough to explain their biological activity so that references is required adding two referenced as below:
Biomedicine & Pharmacotherapy (2009), 63(2), 105-113
Science of the Total Environment (2003), 312(1-3), 15-21.
Author Response
Thank you for the opportunity to respond to the Reviewer’s excellent suggestions for our manuscript titled, “Selenate prevents adipogenesis through induction of selenoprotein S and attenuation of ER stress”. As Reviewer suggested, two references above were cited in the manuscript.
Reviewer 3 Report
Kim et al highlight a new interesting molecular pathway driven by selenate acting to prevent adipocytes differentiation.
The methods are clearly presented and the results are consistent with the proposed hypothesis.
The conclusions are supported by the obtained results.
The Authors should only clarify what do "A", "B", "C" or "AB" mean in figures
Author Response
We really appreciate and agree to reviewer’s valuable opinion. Based on the reviewer’s critical comment, figure legends were revised. The meaning of "A", "B", "C" or "AB" is clarified in the figure legend by describing as “Different characters indicate significant difference at p < 0.05.”